# The Multidisciplinary Approach in the Management of Patients with Kidney Stone Disease—A State-of-the-Art Review

**DOI:** 10.3390/nu16121932

**Published:** 2024-06-18

**Authors:** Krzysztof Balawender, Edyta Łuszczki, Artur Mazur, Justyna Wyszyńska

**Affiliations:** 1Institute of Medical Sciences, Medical College of Rzeszow University, Al. mjr. W. Kopisto 2a, 35-959 Rzeszow, Poland; drmazur@poczta.onet.pl; 2Clinical Department of Urology and Urological Oncology, Municipal Hospital, Rycerska 4, 35-241 Rzeszow, Poland; 3Institute of Health Sciences, Medical College of Rzeszow University, Al. mjr. W. Kopisto 2a, 35-959 Rzeszow, Poland; eluszczki@ur.edu.pl (E.Ł.); jwyszynska@ur.edu.pl (J.W.)

**Keywords:** kidney stones, urinary stone, nephrolithiasis, diet, lifestyle, alcohol, tea, coffee, smoking, physical activity, diabetes, obesity

## Abstract

Kidney stone disease has a multifactorial etiology, and evolving dietary habits necessitate continuous updates on the impact of dietary components on lithogenesis. The relationship between diseases influenced by lifestyle, such as obesity and diabetes, and kidney stone risk underscores the need for comprehensive lifestyle analysis. Effective management of kidney stones requires a multidisciplinary approach, involving collaboration among nutritionists, urologists, nephrologists, and other healthcare professionals to address the complex interactions between diet, lifestyle, and individual susceptibility. Personalized dietary therapy, based on each patient’s unique biochemical and dietary profile, is essential and necessitates comprehensive nutritional assessments. Accurate dietary intake evaluation is best achieved through seven-day, real-time dietary records. Key factors influencing urinary risk include fluid intake, dietary protein, carbohydrates, oxalate, calcium, and sodium chloride. Personalized interventions, such as customized dietary changes based on gut microbiota, may improve stone prevention and recurrence. Current research suggests individualized guidance on alcohol intake and indicates that tea and coffee consumption might protect against urolithiasis. There is potential evidence linking tobacco use and secondhand smoke to increased kidney stone risk. The effects of vitamins and physical activity on kidney stone risk remain unresolved due to mixed evidence. For diseases influenced by lifestyle, conclusive evidence on targeted interventions for nephrolithiasis prevention is lacking, though preliminary research suggests potential benefits. Management strategies emphasize lifestyle modifications to reduce recurrence risks, support rapid recovery, and identify predisposing conditions, highlighting the importance of these changes despite inconclusive data.

## 1. Introduction

Kidney stone disease (KSD) is a pervasive condition affecting around 10–15% of people worldwide, making it the most prevalent among urological disorders [1]. Epidemiological surveys indicate that both the occurrence and prevalence of kidney stones are increasing each year [2,3]. The assessment of risk status for individuals with urolithiasis should be approached comprehensively, encompassing not only the likelihood of recurrence or growth of urinary stones but also considering potential risks for chronic kidney disease (CKD), end-stage kidney disease, and mineral and bone disorder [4,5]. Furthermore, recurrent stones can severely disrupt home and social life, professional activities, personal finances, and emotional and mental health [6,7]. Studies indicate that approximately 50% of those who experience recurrent stone formation encounter only a single recurrence during their lifetime [8]. Recent literature suggests that the rate of recurrence among first-time stone formers is approximately 26% within a five-year interval. Additionally, among individuals who do not engage in prophylaxis, the recurrence rate of secondary deposits is projected at 10–23% annually, reaching 50% within 5–10 years, and escalating to 75% by 20 years following the initial episode [9].

In the urinary tract, stone composition varies considerably, with the majority being calcium-based, constituting approximately 80% of all stones. Uric acid stones account for about 9%, struvite stones make up 10%, and cystine stones represent a rare category, comprising only 1% of cases. Specifically, among calcium stones, which are the most widespread, 50% are primarily made up of calcium oxalate, while 5% consist of calcium phosphate. The remaining 45% are mixed stones, containing both calcium oxalate and phosphate [10]. This diversity in the stone composition is critical for determining the appropriate management and prevention strategies for affected individuals.

The impact of urolithiasis on renal function can be significant, stemming from direct obstructions or infections caused by renal stones, renal tissue damage, or through the urological interventions employed to treat these conditions. An essential aspect in evaluating the consequences of kidney stone disease is the increased risk of kidney cancers. The pooled risk ratio for transitional cell carcinoma (TCC) in individuals with nephrolithiasis is 2.14 (95% CI: 1.35–3.40). For renal cell carcinoma (RCC), the risk in the male population is estimated at 1.41 (95% CI: 1.11–1.80) [11].

In addition to the adverse impacts on patient quality of life, the economic consequences of kidney stones are substantial. It is projected that the yearly expense will surpass $4 billion in the United States by 2030, not including the additional financial burden on patients due to lost work, which averages 19 h per episode per patient [12,13]. British studies indicate that the economic burden linked to KSD was estimated to be between £190 and £324 million in 2010 [14]. This substantial financial impact reflects the direct costs associated with medical treatments and surgeries, as well as indirect costs due to lost workdays and decreased productivity among affected individuals. These findings underscore the significant economic implications of KSD on the healthcare system and the broader economy.

Considering the multifactorial etiology of kidney stone disease, the changing dietary habits necessitate regular updates on the impact of dietary components on lithogenesis. Additionally, there is a need to analyze broadly conceived lifestyle factors and the link between lifestyle diseases like obesity and diabetes and the risk of developing kidney stones. Lifestyle diseases are major public health issues that have been steadily related to the formation of nephrolithiasis. Consequently, KSD is increasingly recognized as a disorder that requires a multidisciplinary approach, both in terms of treatment and prevention. This approach involves integrating dietary management, lifestyle modification, and medical therapies. It highlights the importance of cross-disciplinary collaboration among nutritionists, urologists, nephrologists, and other healthcare professionals to address the intricate interplay between diet, lifestyle, and individual susceptibility to kidney stone formation.

## 2. Research Strategy Employed in the Review of Available Literature

### 2.1. Literature Research

To synthesize data from the literature, comprehensive searches were conducted in March 2024 using databases such as PubMed, Scopus, and Web of Science. The search terms employed included a combination of the following keywords: “Kidney stones”, “Urinary stone”, “Urolithiasis” and “Diet”, “Lifestyle”, “Microbiota”, “Alcohol”, “Tea”, “Coffee”, “Smoking”, “Physical Activity”, “Diabetes”, “Obesity”, “Metabolic Syndrome”. Initial screening involved assessing titles and abstracts, leading to the exclusion of articles that failed to meet the established relevance criteria. Subsequent in-depth analyses were performed on the remaining research and review papers to pinpoint the most pertinent publications. A team of authors reviewed the literature. To maintain consistency in the review process, all reviewers collaboratively screened the identified publications, discussing the outcomes and refining the screening and data extraction protocols prior to commencement.

### 2.2. Selection and Data Extraction

The search parameters were confined to studies published within the period from 2000 to 2024. The most relevant findings were meticulously selected based on their significance. Following an initial review, the authors decided to narrow the scope of the literature analysis. The primary reason for this decision was the numerous contradictory reports published over the past several decades. Limiting the database record search to the period from 2000 to 2024 allowed for the presentation of the most current data regarding the issue under study. Due to the necessity of supplementing the discussion, in the case of one work, the authors made an exception and cited an article published before the year 2000 (citation [15]). After a detailed analysis of the selected literature, 140 articles were ultimately included in the review.

## 3. An Explanation of the Diet for the Kidney Stone Disease

Conservative management is becoming a more popular approach for addressing kidney stones, given that pharmacological treatment involves taking medications for life, with the associated financial cost and inconvenience. Conservative management involves changing one’s diet and lifestyle to lower urine supersaturation, increase crystallization inhibitor excretion, and decrease crystallization promoters. The primary aim of these modifications is to reduce the chances of stone recurrence.

KSD is a multifactorial disorder (Figure 1), with an improper diet being one of the most significant etiological factors. A poor diet can lead to changes in urine chemistry, resulting in low urine pH, and elevated levels of urine calcium, uric acid, and oxalate. Increased levels of urine oxalate may be due to various reasons, including excess oxalate intake, insufficient calcium intake, or endogenous production of oxalate from excessive protein and fructose consumption [16].

Studies related to disease outbreaks have revealed that individuals who adhere to the Dietary Approaches to Stop Hypertension (DASH) dietary routine have a notably reduced chance of developing kidney stones. The diet emphasizes the consumption of fruits, vegetables, nuts, legumes, whole grains, and low-fat dairy and restricts the intake of added sugar, processed meat, and sodium. According to Taylor et al., individuals who adhered to a diet consistent with the DASH diet exhibited a 40–50% lower risk of developing kidney stones [17]. In addition, Maddai et al.’s research demonstrated that adopting a DASH-style diet led to improvements in urine profiles for subjects, including a reduced likelihood of hypocitraturia, hyperoxaluria, and hyperuricosuria [18].

In a study by Borghi et al., individuals with recurrent calcium oxalate kidney stones were randomly assigned to either an intervention diet sufficient in calcium or a control diet low in calcium. After 5 years, there were fewer instances of kidney stones in the intervention group because of reduced levels of urinary oxalate. This decrease was probably a result of the calcium and dietary oxalate’s binding effect in the intestines [19].

### 3.1. Dietary Guidelines for Managing Kidney Stone Disease

To minimize the chances of recurring kidney stone formation, it is advisable to customize the dietary treatment according to the metabolic risk profile of each patient. It is recommended to collect two consecutive 24 h urine samples to identify common metabolic imbalances such as hypercalciuria, hypocitraturia, hyperoxaluria, and hyperuricosuria.

Insufficient consumption of fluids or excessive loss of fluids leading to reduced urine production presents a major risk for the formation of kidney stones [20]. Consequently, individuals employed in professions where water accessibility is limited or those working in elevated temperatures, such as steel mill workers, face a heightened risk of developing kidney stones. Furthermore, individuals with restricted access to fluids and washrooms, such as professional drivers, pilots, or teachers, belong to a high-risk demographic [21]. Physicians working in operating rooms have a higher prevalence of nephrolithiasis (17.4% vs. 9.7%), and they report higher stress levels and lower fluid intake compared to those working at other locations [22]. The importance of consuming fluids in preventing the recurrence of urolithiasis has been thoroughly investigated in several systematic reviews and meta-analyses. They have concluded that maintaining a high total fluid intake and aiming for a urine volume of greater than 2.0 to 2.5 L/day can reduce the risk of stone recurrence [23,24,25]. It is important to consume enough fluids to prevent the recurrence of kidney stones, regardless of the composition of the urine stones and individual risk factors for stone formation [20]. For urolithiasis patients, maintaining a minimum urine volume of 2.0 to 2.5 L/24 h is recommended. Patients with cystinuria require a urine volume of 3.0 L/24 h to reduce urinary cystine concentration below the solubility limit of 1.3 mmol/L at pH 6.0 and prevent recurrence [26]. Research involving 27 individuals with cystinuria revealed that consistently maintaining a daily urine volume of more than 3.0 L significantly decreased the likelihood of recurring stone formation [27].

The levels of calcium and magnesium, which are divalent cations, vary greatly in tap water and drinking water across different geographic areas within a single country [28]. Tap water can be considered hard when its calcium carbonate concentration exceeds 120 mg/L [29], and its consumption can contribute to daily dietary calcium intake. Bicarbonate is naturally present in mineral water and other ions and can increase the body’s buffering capacity. It is a strong alkalizing agent supporting alkalinization therapy and increasing urine pH and citrate excretion [30]. A study on healthy individuals found that drinking 2 L/day of bicarbonate-rich mineral water or consuming 2.55 g/day of potassium citrate increased urine pH and citrate excretion and decreased oxalate excretion. This led to a notable reduction in the comparative oversaturation of calcium oxalate and uric acid in both sets [31].

A double-blind cross-over study was conducted on 34 patients with recurrent calcium oxalate stones to examine the effect of 1.5 L/day of a mineral water rich in bicarbonate (2673 mg/L), compared to 1.5 L/day of a mineral water with a low bicarbonate content (98 mg/L), on the risk of urinary stone formation [32]. The intake of the mineral water high in bicarbonate led to a notable rise in urine pH, citrate, and magnesium excretion when compared to the control group. The decrease in the relative oversaturation of calcium oxalate was comparable in both groups. The presence of bicarbonate in mineral water plays a crucial role in its impact on the risk of calcium oxalate and uric acid stone formation [20,33]. A study involving 129 healthy women and men conducted a randomized trial and found that drinking mineral water that contains at least 1800 mg/L bicarbonate and consuming 1.5 to 2.0 L/day may reduce the dietary acid load and decrease net acid excretion [34]. Alkalinizing urine is very useful in uric acid stones but not advisable for calcium oxalate stone formers as calcium oxalate becomes less soluble at higher pHs. A study comparing the composition of bottled water from 10 different European countries found a significant variation in calcium content, with levels ranging up to 581.6 mg/L [35]. The recommended daily intake of calcium ranges from 1000 to 1200 mg and this can be attained by drinking 2 L of water that is rich in calcium. As a result, it is the responsibility of the dietitian to inform patients about the calcium content in water.

Citrus juices (lemon, orange) contain a significant amount of citric acid, which makes them a healthy dietary alternative to alkalizing agents found in medications. The effect of orange juice on urinary risk factors related to kidney stone formation has been researched by several studies, with varying findings. However, three cohort studies confirm that drinking orange juice regularly can reduce the risk of kidney stone formation [36]. According to epidemiological research, grapefruit increased the likelihood of developing kidney stones, whereas consumption of orange juice did not elevate the risk of kidney stone disease [36,37]. The risk of grapefruit juice was not found in smaller prospective clinical studies, and orange juice was shown to have protective effects on urinary parameters [38].

When managing kidney stone disease through dietary therapy, it is important to consider the oxalate concentration in fruit and vegetable juices. Considering the elevated sugar and energy levels and insufficient dietary fiber found in orange juice, it is recommended to prioritize consuming whole fruit and limit daily fruit juice intake to one serving, diluted with water. According to various studies, it has been found that soft drinks, particularly those that are acidified using phosphoric acid, are significantly associated with an increased risk of recurrent stone formation.

The diet typically contains two primary forms of phosphate: organic and inorganic phosphate (Pi). The addition of inorganic phosphate in the form of Pi salts like phosphoric acid, sodium, potassium, or magnesium salts, as well as modified starches, is essential in food processing as additives. Highly processed food items like unflavored pasteurized and sterilized (including ultra-high temperature) milk, infant and follow-on formulas for infants, bread and rolls, and meat products for adolescents and adults are the main food categories containing significant amounts of these food additives [39]. Research has indicated that an overabundance of phosphorus in the urine can, in specific circumstances, result in the development of stones in the human body [40].

#### 3.1.1. Macronutrients

The consumption of a large amount of protein in one’s diet has been linked to possible harmful impacts on urinary risk factors for the formation of stones, as indicated by numerous studies. It has been suggested that the acid load provided by a high protein intake can elevate urinary calcium levels, while also reducing urine pH and citrate excretion [41].

The consumption of a higher amount of acid-forming foods, as measured by the ratio of animal protein to potassium or net acid excretion (NAE), has been linked to an increased risk of developing kidney stones in large-scale observational studies [42] These findings indicate that the balance between alkalizing fruits and vegetables and total protein intake may influence the risk of kidney stone formation. It is widely acknowledged that fruits and vegetables possess significant alkalizing properties, which can effectively counteract the acid produced during the metabolism of ingested protein. Elevated dietary acidity, leading to a decrease in urine pH, poses a risk for various types of kidney stones, especially the most prevalent type, calcium oxalate stones. Higher urine pH levels are associated with increased excretion of stone-inhibiting citrate and greater calcium-binding capacity, as well as reduced urinary calcium excretion [43].

Several factors related to protein intake, metabolism, and unknown causes, such as low or high urine pH, excessive calcium in the urine, reduced citrate levels in the urine, elevated uric acid in the urine, increased oxalate in the urine, as well as additional dietary and environmental factors, like consuming high amounts of salt and having low urine volume, all make it challenging to directly examine “the inherent impact of protein” on the formation of kidney stones. Nevertheless, many of these risk factors can be partially avoided or decreased by making changes to dietary patterns, for example, by increasing the regular consumption of fruits and vegetables that have an alkalizing effect on metabolism.

Numerous studies have examined the fat intake of individuals with kidney stones compared to those without. Some studies have reported that the fat intake of patients with stone and controls is similar [32], while others have found that stone formers have a higher dietary fat intake [44]. Complex mechanisms have been suggested to explain how the dietary fatty acid pattern, particularly the ratio of n-6 to n-3 polyunsaturated fatty acids, may impact the risk of calcium oxalate stone formation. Numerous studies have examined the effectiveness of fish oil intake in managing stone formation through dietary means. It has been discovered that fish oil supplementation can reduce oxalate excretion in healthy individuals [45], and in most cases, it can also lower urinary excretion of calcium and/or oxalate in patients with calcium stones [46].

The consumption of sucrose has been positively correlated with the risk of stone formation in women in several prospective cohort studies but not in men [47,48]. The increase in calcium excretion following an oral glucose load has been ascribed to an increase in the absorption of calcium in the intestine and a decrease in the reabsorption of calcium in the renal tubules [15]. A systematic review and meta-analysis revealed a positive correlation between fructose consumption and the incidence of stone formation [25]. Nonetheless, the underlying mechanisms of this correlation are not well understood. It is postulated that fructose consumption raises the risk of stone formation, in part, through the alteration of urinary excretion of calcium, oxalate, and urine pH, as well as uric acid metabolism [49,50]. Another important issue is the relationship between hyperinsulinemia and urinary calcium excretion in calcium stone formers with idiopathic hypercalciuria.

Yoon et al.’s research found that the rise in urine calcium linked to euglycemic hyperinsulinemia was minimal and did not show statistical significance when comparing individuals with idiopathic hypercalciuria and those without a history of kidney stone formation. Thus, insulin is unlikely to have a significant pathogenic role in idiopathic hypercalciuria [49].

#### 3.1.2. Oxalate

Significant increases in urinary oxalate concentration can lead to a rise in the urinary supersaturation of calcium oxalate [51]. In a study following 134 patients with recurrent calcium oxalate stones, it was found that the primary urinary factor contributing to relapse after a two-year follow-up is the increase in oxalate excretion [52]. The observation was made in a study involving 20 healthy men and women that a controlled oxalate-rich diet (600 mg/day) led to a significant increase in oxalate excretion from 0.354 to 0.542 mmol/24 h, which is an increase of 0.188 mmol/24 h, more than 50%, corresponding to 35% of total urinary oxalate excretion compared to a normal oxalate diet (100 mg/day) [53]. This study also showed that high dietary oxalate intake leads to a significant increase in the supersaturation of calcium oxalate. Nevertheless, a study following a group of people over time found that there was only a slight positive connection between consuming oxalate in the diet and the likelihood of developing kidney stones [54]. Plant foods are the primary source of dietary oxalate. Estimates of dietary oxalate intake vary widely, depending on the consumption of oxalate-rich foods. Therefore, it is crucial to consider the sources of excess dietary oxalate. In the dietary treatment of patients with calcium oxalate stones, a detailed understanding of the oxalate content of foods is essential.

One of the essential responsibilities of a dietitian is to enlighten patients regarding the dietary sources that contain oxalates. High fluid intake is crucial to prevent the recurrence of kidney stone disease, so the patient should be advised about the oxalate content in fluids. Additionally, it should be noted that boiling can cause considerable losses of oxalate into the cooking water. Therefore, food processing and preparation methods are important factors to consider in determining the oxalate content. To avoid oxalate-rich foods and beverages, patients with kidney stones should be advised accordingly.

#### 3.1.3. Calcium

Consuming a well-balanced diet that includes calcium from both dairy and non-dairy sources has been established as a preventive measure against the formation of urinary stones [55]. Restricting calcium intake in one’s diet should be avoided, as it can result in bone loss and increase the absorption and excretion of oxalate in the urine.

The correlation between a low dietary intake of calcium and an elevated risk of developing stones is likely attributed to an increase in oxalate excretion in urine. Oxalate is assimilated throughout the gastrointestinal tract. When calcium and oxalate are consumed together, a compound of calcium and oxalate is formed within the digestive system, inhibiting the absorption of free oxalate and its subsequent excretion in urine. Conversely, a restricted dietary intake of calcium results in an increased availability of free oxalate for absorption in the digestive system, leading to heightened oxalate excretion in urine [56].

Notably, an epidemiological study has revealed an inverse correlation between dietary calcium intake and the risk of stone formation among both men and women [47]. A study that lasted five years and involved 120 men with calcium oxalate stone formation and hypercalciuria found that maintaining a normal calcium intake of 1200 mg/day was more successful in reducing recurrences than following a low-calcium diet of 400 mg/day [19]. Individuals with idiopathic calcium stone formation are advised to consume a total dietary calcium intake of 1000 to 1200 mg/day.

#### 3.1.4. Sodium Chloride

Consuming dietary sodium chloride can increase the risk of stone formation as it promotes the excretion of urinary calcium. High intake of sodium chloride can cause the excretion of calcium by inhibiting the reabsorption of calcium in renal tubules due to the expansion of extracellular fluid volume caused by sodium [57]. Studies have shown that the daily calcium excretion increases by approximately 1 mmol for every 100 mmol (2300 mg) increase in sodium intake per day in normal adults [58]. A study involving 210 individuals with idiopathic calcium oxalate stones found that a low-salt diet was effective in reducing urinary calcium excretion compared to a control diet [59]. It is recommended to consume less than 100 mmol (2300 mg) or 6 g of salt (sodium chloride) per day.

### 3.2. Microbiota Involvement in Kidney Stone Disease

The relationship between the microbiota, particularly the gut microbiome, represents another emerging aspect considered among the etiological factors for kidney stone formation. *Oxalobacter formigenes*, an anaerobic bacterium, is believed to be a crucial regulator of oxalate in the human body—it uses this molecule as its sole carbon source. Some research indicates that intestinal colonization by this bacterium may be linked to reduced levels of urinary oxalate and, consequently, a decreased risk of developing calcium oxalate stones [60]. It has been observed that there is a notable inverse relationship between the presence of *Oxalobacter formigenes* in the gut and the levels of oxalate found in urine [61]. This process effectively reduces the intestinal absorption and subsequent urinary excretion of oxalate. As a result, the presence of *O. formigenes* diminishes the oxalate available for absorption in the gut [62]. However, while this bacterium impacts oxalate levels, it does not influence other urinary factors associated with the formation of kidney stones, such as the concentrations of calcium, uric acid, sodium, citrate, and magnesium. Moreover, this bacterium is not suitable as a probiotic due to its sensitivity to antibiotics and its intolerance to low pH levels. Consequently, further studies have focused on finding bacteria that can degrade oxalate, concentrating on well-established probiotics like Lactobacillus and Bifidobacterium strains, *Eubacterium lentum*, *Enterococcus faecalis*, and *Escherichia coli*. Research has confirmed a characteristic imbalance in the gut microbiota of patients with kidney stones in comparison to healthy individuals, as evidenced by a meta-analysis that showed these patients had significantly higher levels of Bacteroides (35.11% vs. 21.25%, *p* = 0.0004) and Escherichia_Shigella (4.39% vs. 1.78%, *p* = 0.001) and lower levels of Prevotella (8.41% vs. 10.65%, *p* < 0.00001) [1]. Recent studies suggest that personalized treatments, such as microbial supplements, probiotics or synbiotics, and tailored dietary modifications based on an individual’s specific gut microbiota, might offer improved prevention of stone formation and recurrence.

## 4. Modifiable Lifestyle Factors and Habits

### 4.1. Alcohol Intake

Alcohol remains one of the most widely consumed psychoactive substances worldwide. Previous research has produced inconsistent results concerning the relationship between alcohol consumption and urolithiasis. The differential effects of alcohol type and consumption patterns highlight the importance of considering both quantity and type of alcohol when evaluating kidney stone risk. The protective effect is hypothesized to be due to increased urine volume from the diuretic effect of alcohol and the presence of phytochemicals in some beverages that may aid in preventing crystal formation. Conversely, excessive alcohol intake appeared to increase the risk, possibly due to dehydration and metabolic disturbances. In 2023, the cross-sectional study explored the relationship between alcohol intake and the prevalence of kidney stones by examining the National Health and Nutrition Examination Survey (NHANES) dataset. The findings showed no significant correlation between alcohol consumption, both lifetime and recent (within the past 12 months), and a history of kidney stone formation. Furthermore, the analysis revealed that the quantity and frequency of alcohol consumption did not correlate significantly with the occurrence of kidney stones, even among individuals who consume alcohol heavily [63]. However, results based on Mendelian randomization indicated a possible causal link between the frequency of alcohol intake and the risk of urolithiasis but not with overall alcohol intake [64]. The findings indicated that the frequency of alcohol intake was significantly linked to a higher risk of urolithiasis (odds ratio (OR) 95% confidence interval (CI): 1.31 (1.02, 1.68), *p* = 0.032). However, overall alcohol consumption did not significantly affect the risk of urolithiasis (OR (95% CI): 0.74 (0.48, 1.14), *p* = 0.173).

On the other hand, Kim et al. reported findings from a study involving over 28,000 Korean patients, using data from the Korean National Health Insurance Service–Health Screening Cohort. Alcohol consumption was associated with a decreased likelihood of KSD (adjusted odds ratio (aOR) = 0.89, 95% confidence interval (CI): 0.86–0.92, *p* < 0.001). This inverse association was particularly pronounced in individuals younger than 55 years (aOR = 0.82, 95% CI: 0.78–0.87) and in males (aOR = 0.86, 95% CI: 0.84–0.89) [65]. Similar trends are indicated by the results of a prospective study involving over half a million participants from China. The relationship between alcohol consumption and the risk of kidney stones was examined in a multivariable analysis. The hazard ratio (HR), adjusted for multiple variables, for participants consuming 30.0–59.9 g of pure alcohol per day was 0.79 (95% CI: 0.72–0.87) compared to participants who reported no alcohol consumption in the preceding year. This decreased risk of kidney stones was consistently noted in both weekly and daily drinkers of strong spirits [66]. In addition to data concerning the Asian population, we also have results from European studies indicating a positive correlation between alcohol consumption and a reduced risk of kidney stones. This relationship was analyzed among 51,336 participants who were part of the Oxford cohort in the European Prospective Investigation into Cancer and Nutrition (EPIC). Alcohol consumption was similarly linked to a decreased risk, with the hazard ratio (HR) for the lowest third of intake compared to the highest being 0.65 (95% CI: 0.47–0.91; *p*-value for trend = 0.04) [67]. The epidemiological study results are complemented by the meta-analysis findings presented by Wang et al. According to the study, the pooled odds ratio (OR) estimates suggested that alcohol consumption was associated with a reduced risk of urolithiasis, with an OR of 0.683 (95% CI: 0.577–0.808). Furthermore, the dose–response meta-analysis revealed that the incidence of urolithiasis decreased by 10% for every 10 g per day increase in alcohol consumption, as evidenced by an OR of 0.898 (95% CI: 0.851–0.948) [68].

The underlying mechanisms linking alcohol consumption to the risk of developing kidney stones are still not fully understood. Some studies propose that the protective impact of alcohol against urolithiasis may be attributed to its diuretic properties [65]. Alcohol consumption has been hypothesized to dilute metabolites in the blood and urine, which potentially reduces the risk of forming stones [68]. Furthermore, alcohol may inhibit the secretion of vasopressin, a hormone that regulates urine concentration and volume, thereby possibly preventing the aggregation of minerals that form kidney stones.

Alcohol consumption exhibits a dual relationship with kidney stone risk, mediated by the type and quantity of alcohol consumed. Moderate consumption of beer and wine may offer a protective effect against kidney stone formation, likely due to their high water content and phytochemical composition. In contrast, high consumption, particularly of spirits, may increase the risk. These findings suggest that recommendations for alcohol consumption in the context of kidney stone prevention should be tailored, taking into account individual risk factors and drinking patterns.

### 4.2. Consumption of Coffee and Tea

It is well understood that consuming caffeine has a diuretic effect. However, the connection between caffeine intake and the prevalence of KSD has been previously contested. In a case–control study, Massey et al. noted a positive correlation between caffeine intake and the formation of calcium oxalate stones [69]. Additionally, another cross-sectional study indicated that caffeine consumption was linearly associated with a higher risk of recurrent kidney stones. The multivariate-adjusted odds ratios (with 95% confidence intervals) for recurrent kidney stones increased by 1.15 (ranging from 1.01 to 1.31) for each quartile increase in caffeine intake [70].

On the other hand, the UK Biobank study found that genetically predicted coffee and caffeine intake was linked to a reduced risk of KSD. Based on the Zhong et al. study, single-nucleotide polymorphisms (SNPs) associated with coffee consumption at the genome-wide significance level were obtained from a meta-analysis of four genome-wide association studies (GWAS). Of the 17 replicated loci in the study, 5 have been associated with coffee consumption or plasma caffeine metabolites (GCKR, ABCG2, AHR, POR, and CYP1A1/2) [71]. The combined odds ratio for kidney stones was 0.60 (95% CI: 0.46–0.79; *p* < 0.001) for every 50% increase in coffee consumption predicted by genetics and 0.81 (95% CI: 0.69–0.94; *p* = 0.005) for every 80 mg increase in caffeine consumption predicted by genetics [72]. In a comprehensive meta-analysis, which reviewed seven studies totaling 772,290 cohort members and 9707 kidney stone cases, evidence emerged of a significant inverse relationship between caffeine consumption and the risk of kidney stones. Specifically, those in the highest category of caffeine intake exhibited a 32% reduction in risk (RR = 0.68; 95% CI: 0.61–0.75) compared to those with the lowest intake. This association persisted consistently across various subgroups, suggesting a protective effect of caffeine against the formation of kidney stones [73].

Several mechanisms support the hypothesis that increased caffeine consumption is linked to a lower risk of developing kidney stones. Caffeine significantly affects the activity of the antidiuretic hormone, which leads to an increase in urine flow and a decrease in its maximum concentration. Additionally, caffeine inhibits the formation of calcium oxalate stones and reduces the adhesion of calcium oxalate crystals to the renal tubular epithelial cells’ surfaces [73]. In another study, Peerapen and Thongboonkerd provided in vitro evidence of caffeine’s protective mechanism against kidney stone formation. They demonstrated that caffeine facilitates the translocation of annexin A1 from the apical surface into the cytoplasm, reducing the crystal-binding capacity of renal tubular epithelial cells [74].

It is also worth mentioning the consumption of decaffeinated coffee in the context of kidney stone risk. Decaffeinated coffee also appears to serve a protective role against stone formation [75]. The main explanations for this observation include the presence of small amounts of caffeine in decaffeinated coffee, which may still be active. Additionally, coffee contains other protective bioactive compounds such as trigonelline, which may have protective effects similar to caffeine. The content of trigonelline in coffee varies depending on the roasting and brewing methods, as well as the type of coffee used [76].

The assessment of the relationship between tea consumption and the risk of urinary stones presents another challenging issue. On the one hand, tea contains oxalates, which could potentially increase the risk of developing kidney stones. However, on the other hand, tea is abundant in polyphenols and various other phytochemicals, and the antioxidant properties of these components offer protection against urinary stone formation [77]. Furthermore, the caffeine present in tea may reduce the adherence of calcium oxalate crystals to the renal tubular epithelial cells [74].

Liu et al. carried out a Mendelian randomization study to determine if the link between tea consumption and the risk of urolithiasis is causal. In a robust analysis using genome-wide association study data sourced from the FinnGen consortium and UK Biobank, encompassing a total of 12,521 cases, authors applied inverse variance-weighted analysis techniques. The findings revealed a significant causal relationship between genetically predicted tea intake and a lower incidence of kidney stones. Similar to coffee consumption, genetic associations for tea intake were taken from a large genome-wide association study (GWAS), conducted in 448,060 individuals of the UK Biobank (44 independent genetic variants strongly associated with tea consumption were identified). Specifically, the odds of developing kidney stones were reduced by 53% for individuals with genetic markers indicating higher tea consumption, as evidenced by an odds ratio of 0.47 (95% CI: 0.34–0.66; *p* < 0.001). This study provides compelling genetic evidence supporting the protective effects of tea against the formation of kidney stones [78]. In a comprehensive observational study involving three distinct cohorts and a cumulative participant count of 194,095, Ferraro et al. explored the relationship between tea consumption and the incidence of kidney stones. The analysis revealed a statistically significant protective effect associated with regular tea consumption. Specifically, individuals who consumed at least one glass of tea daily were found to have a decreased risk of kidney stone formation compared to those who consumed less than one glass per week. The hazard ratio was determined to be 0.89, with a confidence interval of 0.82 to 0.97, indicating a robust inverse association between tea intake and the likelihood of developing kidney stones [36]. When studying tea consumption, it is crucial to distinctly assess the impact of green tea intake on the development of kidney stones. Green tea contains a specific bioactive polyphenol compound known as epigallocatechin gallate [79]. An in vitro study found that epigallocatechin gallate inhibits the oxalate-induced translocation of α-enolase, another protein that binds to calcium oxalate monohydrate (COM), thereby reducing the COM-binding capacity of renal epithelial cells [80]. Shu et al. investigated the association between green tea intake and incident stones in two large prospective cohorts. Tea consumption was associated with a lower risk of incident events among drinkers, with men showing a hazard ratio of 0.78 (95% CI: 0.69–0.88) and women a hazard ratio of 0.80 (95% CI: 0.77–0.98). Specifically, green tea consumers also exhibited reduced risks, with hazard ratios of 0.78 (95% CI: 0.69–0.88) for men and 0.84 (95% CI: 0.74–0.95) for women, compared to those who never drank or were former drinkers [81].

Despite concerns regarding the oxalate content in tea and its potential to contribute to kidney stone formation, a comprehensive review of clinical studies, particularly those involving large cohorts, suggests a different narrative. Analysis of these studies indicates that the overall evidence predominantly supports the protective effects of tea consumption against the development of urolithiasis.

### 4.3. Smoking

Recent studies have indicated a potential link between tobacco use, including exposure to secondhand smoke, and the increased risk of developing KSD. The underlying hypothesis is that tobacco smoke introduces chemicals into the body that exacerbate oxidative stress and elevate levels of vasopressin. This hormonal increase can lead to reduced urine output, a critical factor in kidney stone formation due to the concentrated minerals in the urine.

Based on the National Health and Nutrition Examination Survey (NHANES) database study, current smokers exhibited a significantly higher risk of developing kidney stones compared to non-smokers, with an odds ratio of 1.17 (95% CI: 1.04–1.31, *p* = 0.009), and a positive trend in risk was also noted. Additionally, individuals with serum cotinine levels ranging from 0.05 to 2.99 ng/mL showed an increased risk of nephrolithiasis with an OR of 1.15 (95% CI: 1.03–1.29, *p* = 0.013) [82]. The present research also indicates a significant link between passive smoking and an increased risk of kidney stone formation. The incidence of kidney stones was found to be significantly higher in the secondhand smoker group compared to the non-exposure group, with an odds ratio of 1.64 (95% CI: 1.21 to 2.23). Furthermore, participants exposed for more than 1.2 h per week were nearly twice as likely to develop kidney stones as those with no exposure, exhibiting an OR of 1.92 (95% CI: 1.29 to 2.86) [83].

### 4.4. Vitamins Supplementation

Much of the research exploring the potential role of vitamins in kidney stone formation has concentrated on the effects of vitamin C and vitamin B6, both involved in oxalate metabolism. Vitamin C can be converted into oxalate, whereas vitamin B6 acts as a cofactor in the metabolism of oxalate.

According to an earlier study, consuming 1000 mg or more of vitamin C per day was associated with a 41% increased risk (95% CI: 11%–80%) of developing a first kidney stone [48]. Additionally, a separate study involving more than 23,000 Swedish men indicated an increased risk of kidney stones with the use of supplemental vitamin C, showing a multivariable-adjusted hazard ratio of 1.92 (95% CI: 1.33–2.77) compared to non-users [84].

However, based on findings from more recent studies, a higher intake of vitamin C was inversely associated with the risk of stone formation. Specifically, daily intake levels between 60 and 110 mg were linked to a reduced risk (OR = 0.76; 95% CI: 0.60–0.95), as were levels above 110 mg (OR = 0.80; 95% CI: 0.66–0.97) [85]. In a comprehensive cross-sectional study, Zeng et al. explored the relationship between nine common vitamins and the prevalence of kidney stones. Their regression analysis revealed that, compared to lower intakes, high intakes of vitamin B6 (OR (95% CI): 0.76 (0.62, 0.93)), vitamin C (OR (95% CI): 0.73 (0.59, 0.90)), and vitamin D (OR (95% CI): 0.77 (0.64, 0.94)) demonstrated protective effects against kidney stones. Additionally, the relationship between vitamin C intake and kidney stone prevalence exhibited a J-shaped curve, initially decreasing before increasing again [86].

The unambiguous explanation of such conflicting results still poses many challenges. Previous studies have suggested that the primary mechanism through which vitamin C influences the formation of kidney stones is by enhancing the metabolism of oxalic acid. An in vivo study has demonstrated that vitamin C can elevate the excretion of urinary oxalate in hyperoxaluric rats induced with hydroxy-L-proline [87]. Some researchers speculate that the pro-lithogenic association between vitamin C and kidney stones could be attributed to inadequate preservation techniques or abnormal pH levels during the urine preservation, detection, and analysis processes [86,88]. In contrast, an in vitro study has demonstrated the antioxidative properties of vitamin C, showcasing a protective impact against oxalate-induced oxidative stress and renal damage [89]. Furthermore, another in vitro study has proposed that vitamin C hinders struvite crystallization in the presence of *Pseudomonas aeruginosa* [90].

### 4.5. Physical Activity

Recent studies have investigated the potential protective role of physical activity against KSD, but the findings remain inconclusive. A comprehensive cohort study involving nearly 90,000 postmenopausal women indicated that physical activity, irrespective of intensity, might reduce the risk of KSD [91]. This association was further supported by a meta-analysis encompassing over 200,000 participants, which linked higher levels of physical activity with a lower incidence of kidney stones in women [92]. In another study reported by Feng et al., physical activity shows an inverse relationship with the prevalence of kidney stones, exhibiting a dose–response effect that plateaus beyond a certain level of activity. Specifically, as physical activity increases, the prevalence of kidney stones decreases until reaching a plateau at about 2480 MET-minutes per week, where the odds ratio (OR) stabilizes at 0.75 (95% CI: 0.63–0.91), indicating no further reduction in kidney stone prevalence with additional physical activity [93].

However, results from subsequent meta-analyses have failed to establish a significant connection between physical activity and the risk of developing kidney stones, suggesting that earlier positive findings may have been affected by various confounding factors [94]. Given the mixed evidence and the presence of confounding variables in these studies, the actual impact of physical activity on KSD risk remains controversial. This underscores the need for further research, particularly studies that adjust for multiple variables to clarify the true effects of physical activity on kidney stone prevention. Such investigations are essential to establish clear guidelines and potentially preventive measures against KSD.

## 5. Disease Influenced by Lifestyle Related to Kidney Stone Disease

Nephrolithiasis is now recognized as a complex disease that is interconnected with various systemic conditions. Studies have shown links between nephrolithiasis and conditions such as obesity, diabetes mellitus (DM), and metabolic syndrome (MetS) [95]. Individuals with kidney stones are more likely to be affected by a range of comorbidities, including obesity, DM, and MetS, which are also recognized as risk factors for the development of kidney stones [96].

### 5.1. Obesity

Obesity is a complex and multifactorial condition that affects people of all ages and is associated with various health problems. It is influenced by multiple factors, including genetics, socioeconomic status, race, culture, and social environment. Obesity rates have risen globally in recent years [25]. Body mass index (BMI) is commonly utilized in the assessment of overweight and obese individuals in epidemiological investigations [97]. Furthermore, a high BMI is associated with a higher risk of developing severe health conditions, including cardiovascular disease, hypertension, stroke, cancer, DM, and musculoskeletal disorders, which can significantly impact an individual’s overall well-being and life expectancy [98,99]. Other studies suggest that an elevated BMI relates to abnormalities in urine chemistry, indicating potential kidney stone formation, variations in urinary sodium–creatinine ratios, and alterations in calcium, sodium, oxalate, magnesium, phosphate, and citrate levels in urine. Moreover, changes in urine pH led to hypercalciuria [100,101,102]. Understanding the importance of obesity in relation to the development of kidney stones is crucial, considering the common co-occurrence of diseases related to a high BMI alongside kidney stones [103]. This interconnection extends to conditions such as DM, cardiovascular diseases, and hypertension, highlighting the need for comprehensive knowledge in this area.

A recent systematic review and meta-analysis by Emami et al. investigated the correlation between obesity and kidney stone risk. Fifteen observational studies conducted between 2005 and 2022 were included, with findings indicating that individuals with obesity had a 1.35 (95% CI: 1.20 to 1.52, *p* < 0.001) times higher risk of developing kidney stones compared to non-obese individuals. Geographically, the risk among individuals with obesity was significantly higher in North America and Europe compared to Asia [104].

Another meta-analysis by Wang et al. analyzed the association between BMI and kidney stones. The analysis pooled data from 15 studies, involving over 13,000 patients. Patients with urolithiasis were divided into two groups: BMI < 25 and BMI ≥ 25 kg/m^2^. The results did not show a significant association between BMI and urinary stone size. However, excess weight elevates the risk of uric acid stones across genders and regions. Moreover, an increased risk of calcium oxalate stone formation was observed in the excess weight group across all patients. Conversely, in this meta-analysis, no significant association between BMI and calcium phosphate was indicated [105].

A systematic review and dose–response meta-analysis by Taheri et al. assess the impact of BMI on the urinary excretion of different metabolites in patients with kidney stones. The analysis differentiated between normal weight (BMI < 25 kg/m^2^) and excess weight (BMI ≥ 25 kg/m^2^) patients. Normal-weight patients excreted lower levels of calcium, uric acid, oxalate, sodium, citrate, and magnesium but had a higher urinary pH. Moreover, it was indicated a linear dose–response association between BMI and the 24 h excretion of oxalate, uric acid, sodium, phosphate, citrate, and creatinine, indicating that as BMI increases, so does the excretion of these substances. Individuals with excess weight are at a higher risk of forming kidney stones, as they excrete more stone-promoting substances (calcium, sodium, oxalate, and uric acid) as well as stone inhibitors (citrate and magnesium), even though the urinary pH and prevalence of kidney stones remain higher in these groups [106]. The higher urinary excretion of stone-forming substances in obese patients is probably due to a combination of factors. The presence of obesity is associated with metabolic abnormalities, lower urine pH, and increased urinary calcium and oxalate excretion, contributing to stone formation [107,108]. In addition, obesity is associated with insulin resistance and compensatory hyperinsulinemia, which may be a risk factor for the formation of calcium-containing kidney stones. Insulin resistance is associated with defects in renal ammonium production [108].

Larger body size may lead to increased urinary excretion of uric acid and oxalate, which are both risk factors for developing calcium oxalate kidney stones. Urinary oxalate excretion increases with increasing lean body mass, likely due to changes in endogenous oxalate synthesis [48].

In a systematic review and meta-analysis examining the link between body fat, diabetes, physical activity, and the risk of KSD, data from 13 cohort studies revealed that increases in BMI, waist circumference, and weight are linked to an increased risk of nephrolithiasis. Specifically, each 5 unit increase in BMI, 10 cm increase in waist circumference, and 5 kg increase in weight or weight gain were linked to a higher relative risk of developing kidney stones. Additionally, people with DM were found to have a higher risk compared to those without diabetes. Conversely, no significant correlation was found between levels of physical activity and the risk of developing kidney stones [94].

The above meta-analyses underline the importance of implementing mitigation strategies, such as community health programs, to tackle the increased risk of KSD linked to obesity. They also emphasize the potential significance of weight loss in the management of nephrolithiasis. With the rising prevalence of obesity, it is crucial for physicians to inform their patients with overweight or obese about the associated risks to prevent complications from these conditions. A lifestyle modification is advised for these patients, which includes incorporating a suitable diet and engaging in higher levels of physical activity, as outlined in the corresponding sections of this article.

### 5.2. Diabetes Mellitus and Metabolic Syndrome

Individuals diagnosed with both DM and MetS are at an increased likelihood of developing kidney stones [109,110]. The worldwide incidence of these conditions has escalated to levels of a pandemic, increasing concurrently with kidney stone disease [111,112,113]. There is a link between the two conditions, with impaired glucose tolerance being a component of the MetS [114]. While the exact cause of kidney stone disease is still uncertain, individuals with MetS or DM have been found to have higher levels of urinary acidification and production of uric acid stones compared to those without these conditions. Interestingly, as BMI increases in both diabetic and non-diabetic patients, the occurrence of uric acid stones tends to increase while the occurrence of calcium oxalate stones decreases [105,115].

Diabetes mellitus is a chronic, complex metabolic disorder with heterogeneous pathogenesis. This disorder is characterized by hyperglycemia, or elevated blood glucose levels, which results from abnormalities in insulin secretion or insulin action, or both [116]. In 2019, it was estimated that the global prevalence of DM was 9.3%. This figure is expected to increase to 10.2% by 2030 and to 10.9% by 2045. The prevalence is greater in urban areas, at 10.8%, compared to rural areas, at 7.2%, and in high-income countries, at 10.4%, compared to low-income countries, at 4.0%. Furthermore, it is estimated that half of the individuals who have diabetes are unaware that they have the condition [111]. Evidence showed strong connections between DM and various health conditions such as cognitive decline, functional impairment, affective disorders, sleep apnea, liver disease, and infection [117]. Individuals with DM are at a higher risk for kidney stones due to various factors, including changes in urine composition, increased oxalate excretion, and decreased citrate levels. Moreover, those with DM often have additional risk factors like obesity and hypertension, which further increase their susceptibility to kidney stones [118]. A systematic review and meta-analysis by Geraghty et al. investigated the relationship between chronic hyperglycemia in the form of DM and impaired glucose tolerance in the context of MetS and kidney stone disease. A total of 13 studies were included in the meta-analysis. The risk of developing kidney stone disease was found to increase with chronic hyperglycemia, particularly in patients with DM and MetS. This may lead to a higher burden of morbidity and mortality related to kidney stones, especially considering the global rise in DM cases [119]. Another systematic review and Bayesian meta-analysis by Rahman et al. assessed the relationship between MetS, its individual components, and the occurrence of nephrolithiasis. Findings from an analysis of 25 studies involving a total of 934,588 participants indicate that individuals with MetS are 1.769 times more likely to develop nephrolithiasis compared to those without metabolic syndrome. Specifically, the summary ORs for hypertension and dyslipidemia in relation to nephrolithiasis were 1.613 and 1.586, respectively. Furthermore, the analysis showed that DM and obesity are linked to nephrolithiasis with ORs of 1.552 and 1.531, respectively [120]. Yuan and Larsson conducted a Mendelian randomization study to investigate the potential causal relationship between obesity, type 2 DM, and kidney stones. By reviewing data from comprehensive genetic studies, it was found that higher genetically predicted BMI and genetic risk for type 2 DM were linked to a higher risk of nephrolithiasis [121].

### 5.3. Management of Kidney Stone in Relation to Obesity, Diabetes Mellitus, and Metabolic Syndrome

Effective treatment of nephrolithiasis involves a combination of behavioral and nutritional interventions that also target diseases that increase the risk of KSD, together with tailored pharmacological treatment that is specific to the type of kidney stone.

The current guidelines from the American Urological Association emphasize investigating the causes of nephrolithiasis in affected individuals but do not specify treating obesity, DM, or MetS as part of medical management [26]. The efficacy of lifestyle modifications, including caloric restriction and increased physical activity, in mitigating nephrolithiasis frequency and enhancing treatment outcomes remains underexplored. Nonetheless, such interventions are linked to decreased cardiometabolic risks in MetS and DM, potentially benefiting nephrolithiasis management. While further studies are needed for definitive therapeutic recommendations, it seems reasonable and justifiable to recommend these lifestyle changes as adjunctive treatments for nephrolithiasis complicated by MetS, in alignment with the standard management of nephrolithiasis [122].

The existing evidence does not provide a clear understanding of whether targeted interventions can effectively reduce the increased risk of kidney stones, specifically uric acid nephrolithiasis, in individuals with hypertension, obesity, diabetes mellitus (DM), and metabolic syndrome (MetS)—all representing insulin-resistant conditions [122,123]. However, findings from a basic research model indicate a potential positive impact. Sasaki et al.’s research on a rat model examined the impact of caloric restriction and physical activity on urinary stone formation, revealing that body mass reduction decreased the risk of uric acid and calcium oxalate nephrolithiasis by enhancing urine pH and urinary citrate excretion. This constitutes preliminary evidence that weight management and exercise may decrease nephrolithiasis risk, warranting further investigation [124]. There are also proposals suggesting that promoting dietary habits and physical activity may be beneficial for individuals with MetS to prevent kidney stone formation. Obesity and overweight are identified as risk factors for KSD; therefore, the European Association of Urology and Urological Association of Asia recommends weight reduction to achieve and maintain a healthy BMI as part of strategies to lower the risk of nephrolithiasis [125,126].

## 6. Primary Hyperparathyroidism and Risk of Calcium Stone Disease

Although currently the predominant form of primary hyperparathyroidism (PHPT) has changed from symptomatic to asymptomatic disorders worldwide, and the prevalence of PHPT among patients with nephrolithiasis is relatively low [127], parathyroid diseases should always be considered in the diagnosis of kidney stones, especially in its recurrent form.

PHPT is an endocrine disorder characterized by the excessive secretion of parathyroid hormone, which primarily affects the skeletal system and kidneys. This hyperfunctioning of the parathyroid glands leads to an imbalance of calcium homeostasis, resulting in various clinical manifestations, including osteitis fibrosa cystica, osteoporosis, and nephrolithiasis [128,129]. The development of nephrolithiasis is a recognized initial presentation of PHPT, with 7–20% of patients eventually experiencing kidney stone formation [130]. Historically, primary hyperparathyroidism was often diagnosed in patients who exhibited symptoms like kidney stones, bone pain, fractures, muscle weakness, or bone deformities. However, since the 1970s, most cases in developed countries have been identified through routine blood tests, which revealed elevated calcium levels in patients who were otherwise asymptomatic [131]. When evaluating patients, it is essential to take a comprehensive medical history, inquiring about factors that may contribute to hypercalcemia, such as a history of kidney stones, bone pain, muscle weakness, depression, and use of medications like thiazide diuretics, calcium-containing products, and vitamin D supplements [132]. Primary hyperparathyroidism often presents without symptoms, but 55% of patients may exhibit undiagnosed nephrocalcinosis or non-obstructing renal calculi, while 75% of symptomatic patients experience acute renal colic or nephrolithiasis [133]. Serum calcium levels should be monitored in all patients with calcium nephrolithiasis.

The timely detection and treatment of PHPT is crucial for preventing recurrent stone disease. Surgery is the only definitive treatment [134]. In accordance with the American Association of Endocrine Surgeons’ guidelines, surgery is recommended for individuals with symptomatic PHPT characterized by nephrolithiasis, fragility fractures, and osteoporosis, as well as asymptomatic patients with elevated serum calcium levels exceeding 1 mg/dL above normal [135,136]. Evidence suggests that timely surgical intervention is crucial for the management of hyperparathyroidism in both symptomatic and asymptomatic patients, as prolonged uncontrolled PHPT can lead to serious complications, including recurrent stone disease [137,138].

Following successful parathyroidectomy, serum calcium levels normalize permanently, and parathyroid hormone levels decrease, leading to significant improvements in bone mineral density, microstructure, and strength. This results in a reduced risk of fractures and kidney stones, as well as enhanced overall skeletal health [139,140].

## 7. Summary of Multidisciplinary Evidence-Based Protective Strategies for KSD

Dietary management is important in the prevention and treatment of kidney stones. The type of kidney stone will determine the specific dietary modifications needed. The comprehensive management of a patient presenting with nephrolithiasis involves the active engagement of a multidisciplinary team, which includes a registered dietitian (Table 1).

The care provided by a dietitian to a patient is a holistic approach that involves understanding the underlying causes of their condition. This includes analyzing any environmental and lifestyle-related factors that may be contributing to the ailment. Once the factors are identified, the dietitian devises a comprehensive nutritional plan that is tailored to the patient’s unique needs, preferences, and health goals.

The nutritional plan may include a variety of dietary modifications, such as increasing or decreasing the intake of certain nutrients, incorporating specific foods or supplements, and adjusting the timing and frequency of meals. The goal of this plan is not only to enhance the patient’s quality of life but also to reduce healthcare and public health expenses by preventing the ailment from recurring. By empowering patients to make informed decisions about their health, dietitians play a critical role in improving overall health outcomes.

## Figures and Tables

**Figure 1 nutrients-16-01932-f001:**
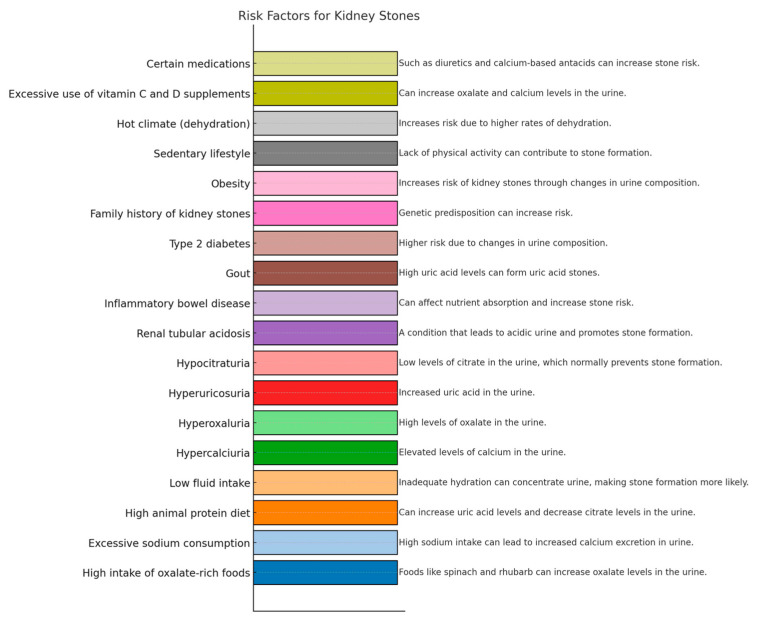
The most important risk factors for kidney stone disease.

**Table 1 nutrients-16-01932-t001:** Multidisciplinary evidence-based protective strategies for KSD.

The Role of the Registered Dietitian	Modifications in Lifestyle and Habits	Additional Recommendations for Individuals with Disease Influenced by Lifestyle
Correcting urinary risk factors for kidney stone formation can be achieved through diet modification, particularly in the case of the most common stone type, calcium oxalate.	Current research indicates that guidance on alcohol intake for kidney stone prevention should be personalized, considering individual risk factors and consumption patterns.	Current evidence fails to conclusively determine if targeted interventions can effectively lower the risk of nephrolithiasis in individuals with hypertension, obesity, diabetes mellitus, and metabolic syndrome. Nevertheless, initial findings from basic research models suggest a potential beneficial impact.
To ensure successful dietary therapy for stone-forming patients, dietary therapy should be individualized based on each patient’s unique biochemical and dietary risk profile.	Analysis of up-to-date studies mostly reveals that tea consumption tends to have protective effects against the development of urolithiasis.	Management of nephrolithiasis emphasizes lifestyle modifications to mitigate recurrence risks, alongside rapid recovery, and identification of predisposing systemic conditions. It is recommended to emphasize lifestyle changes even in the absence of clear-cut data on their effects on kidney stone prevention.
A comprehensive nutritional assessment is a crucial aspect of the evaluation and a prerequisite for effective dietary therapy in stone-forming patients.	Recent research has suggested a potential association between tobacco use, including secondhand smoke exposure, and a heightened risk of developing kidney stone disease.	
Habits in dietary intake can be accurately gauged through 7-day dietary records, which are considered the most precise method for evaluating dietary intake.	The relationship between common vitamins and the prevalence of kidney stones remains unresolved according to current research findings.	
The risk of kidney stone formation can be reduced by adjusting the urinary risk profile through different dietary factors. These factors include fluid intake, dietary protein, carbohydrates, oxalate, calcium, and sodium chloride.	Given the mixed evidence and the influence of confounding variables in the existing studies, the actual effect of physical activity on kidney stone disease risk continues to be a subject of debate.	
Personalized interventions, including microbial supplements, probiotics, synbiotics, and customized dietary changes tailored to an individual’s unique gut microbiota, could enhance the prevention of stone formation and its recurrence.

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
