# Peer review of "The Multidisciplinary Approach in the Management of Patients with Kidney Stone Disease—A State-of-the-Art Review"

_nutrients, 2024, doi:10.3390/nu16121932_

Round 1

Reviewer 1 Report

Comments and Suggestions for Authors

Balawender et al wrote review article about kidney stone, especially management of diet.

I think this article is well written and contains sufficient information of daily treatment.

This paper contains a wide range of necessary and sufficient information, from the formation of urinary stones to dietary therapy and underlying diseases.

Among these, we would like to request two additional points, but since they are also related to the content, please consider responding if possible.

1.      It has been mentioned that phosphorus is bad for urinary stones. However, how about mentioning more specifically about foods that contain excess phosphorus? In particular, foods that contain a lot of food additives contain inorganic phosphorus, which is more easily absorbed and is reported to be bad for urinary stones.

2.      One of the diseases that can cause urinary stones is primary hyperparathyroidism. This disease is often discovered as a result of urinary stones, and I don't think it is considered a differential diagnosis unless it is suspected. If a person with a low risk of developing urinary stones develops, why not state that you suspect it?

Comments on the Quality of English Language

n/a

Author Response

RESPONSE TO REVIEWER 1 COMMENTS

We would like to express our appreciation to the reviewers and editorial board for taking the time and effort to improve our work and provide such insightful comments.

We are pleased to have been given the opportunity to revise our manuscript entitled “The multidisciplinary approach in the management of patients with kidney stone disease – a state-of-the-art review”.

We have carefully reviewed your comments. Below we explain how we revised the paper based on your comments and recommendations.

  1. It has been mentioned that phosphorus is bad for urinary stones. However, how about mentioning more specifically about foods that contain excess phosphorus? In particular, foods that contain a lot of food additives contain inorganic phosphorus, which is more easily absorbed and is reported to be bad for urinary stones.

Response: Thank you very much for your comment. This information has been added to the article. Please see lines 231-239.

  1. One of the diseases that can cause urinary stones is primary hyperparathyroidism. This disease is often discovered as a result of urinary stones, and I don't think it is considered a differential diagnosis unless it is suspected. If a person with a low risk of developing urinary stones develops, why not state that you suspect it?

Response: The article has been supplemented with a chapter on primary hyperparathyroidism and its association with the risk of lithogenesis (Chapter 6, lines 701-738).

Additionally, following the recommendation of the editorial board, the article has been enriched with one figure (Figure 1). The most important risk factors for kidney stone disease), and a table (Table1) summarizing the current knowledge regarding recommendations for the treatment and prevention of kidney stones.

We did our best to improve the manuscript, we hope that we have met your expectations.

Thank you.

Reviewer 2 Report

Comments and Suggestions for Authors

This review is comprehensive and provides some very clear recommendations at the end after a review of the literature. There are several broad generalizations in the body of the review, however, that require correction/modification with more critical appraisal of the particular studies quoted. Some of the inferences made from the references are beyond what could be/should be done given the data provided and need to be corrected.

Line 14: It is preferable to avoid characterizing obesity and diabetes as “lifestyle diseases” as we are discovering that there are many factors in addition to “lifestyle” that can influence blood glucose and even obesity. Later in line 27 it may also be less prejudicial to state “diseases influenced by lifestyle” rather than “lifestyle-related diseases.”

Line 20: By “seven day dietary records” we presume this is an ongoing diary in real time, not recall.  Would emphasize this by restating as “seven-day, real time dietary records” or something to that effect.

Line 48:  Exactly what is meant by “metaphylaxis” in this context; is “prophylaxis” intended? Metaphylaxis is a term for treating individuals without a disease to prevent development of a disease (usually with respect to infection).

Section 3. The discussion would benefit from a figure depicting the pathways for fructose and protein catabolic end products leading to increased oxalate.

Lines 134-148 would be better at the end of the discussion re: the components involved in renal stones so that dietary modifications can be better understood in context. At this point, the text is a broad description that could apply to any disorder where diet has an impact does not provide any specific guidance regarding stone disease.

Line 174: Cations does not require an apostrophe.

Line 196: Alkalinizing urine is very useful in uric acid stones but not advisable for calcium oxalate stone formers as calcium oxalate becomes less soluble at higher pHs.

Line 204: The benefit vs risk of grapefruit juice (as opposed to other citrus fruits) should be noted. Some epidemiological studies show increased risk of stones with grapefruit juice ingestion whereas there are other studies that show no change, improvement or worsening in some parameters associated with stone formation (e.g., increased citrate excretion but increased oxalate excretion also). These discrepancies should be noted.

Lines 217ff:  The intake of protein can have disparate effects on stone formation depending on the type of stone (uric acid vs calcium oxalate or phosphate). These should be distinguished. Also L methionine is not a protein  and should be considered distinct from protein itself in this regard.

Lines 242-245:  What is the data regarding oral glucose load and calcium handling by the intestine and kidney tubules? No references are given.

Lines246-249:  Ref. #48 does not address fructose effects on calcium, oxalate and urine pH; the study was a euglycemic insulin clamp.

Lines 278-289:  Any data on the mechanism whereby a normal calcium diet reduces renal stone formation?

Lines 385 – 386: Again, extrapolation that increased calcium excretion (Ref 67) results from decreased tubular reabsorption of calcium without direct data is not accurate. It could well be that alcohol increases calcium filtration (due to lower serum albumin and more free calcium) and thus increases excretion. Important not to extrapolate mechanisms from observations not designed to address the mechanism itself.

Line 403: What is “genetically predicted coffee…consumption”? Is there a gene that makes people drink caffeine (twin studies have been published but the gene is not identified) ? Is caffeine “metabolism” intended via cyp1A2? Likewise “genetically predicted teach intake” (line 443).

Lines 496ff: Any idea why the conflicting findings with vit C?  Please provide a critical appraisal of the studies that may identify potential reasons for the disparate findings as done for physical activity?

Line 536: Unclear why diabetes and obesity are categorized under “disease of affluence”. Poverty also results in diabetes and obesity due to ingestion of cheaper but high calorie, less nutritional foods.  Please recharacterize especially as the subsequent paragraphs actually contradict this heading.

Lines 577ff: Is the higher urinary excretion of stone forming substances in obese patients due to increased intake rather than the obese status itself or other factor?

Author Response

RESPONSE TO REVIEWER  COMMENTS

We would like to express our appreciation to the reviewers and editorial board for taking the time and effort to improve our work and provide such insightful comments.

We are pleased to have been given the opportunity to revise our manuscript entitled “The multidisciplinary approach in the management of patients with kidney stone disease – a state-of-the-art review”.

We have carefully reviewed your comments. Below we explain how we revised the paper based on your comments and recommendations.

This review is comprehensive and provides some very clear recommendations at the end after a review of the literature. There are several broad generalizations in the body of the review, however, that require correction/modification with more critical appraisal of the particular studies quoted. Some of the inferences made from the references are beyond what could be/should be done given the data provided and need to be corrected.

Response: Thank you for your valuable comment and efforts to improve our manuscript.

Line 14: It is preferable to avoid characterizing obesity and diabetes as “lifestyle diseases” as we are discovering that there are many factors in addition to “lifestyle” that can influence blood glucose and even obesity. Later in line 27 it may also be less prejudicial to state “diseases influenced by lifestyle” rather than “lifestyle-related diseases.”

Response: It has been corrected.

Line 20: By “seven day dietary records” we presume this is an ongoing diary in real time, not recall.  Would emphasize this by restating as “seven-day, real time dietary records” or something to that effect.

Response: It has been corrected.

Line 48:  Exactly what is meant by “metaphylaxis” in this context; is “prophylaxis” intended? Metaphylaxis is a term for treating individuals without a disease to prevent development of a disease (usually with respect to infection).

Response: It has been corrected.

Section 3. The discussion would benefit from a figure depicting the pathways for fructose and protein catabolic end products leading to increased oxalate.

Response: Thank you very much for your valuable comment. Due to the extensive nature of the manuscript and the addition of a figure and table to complement the text (version after the first round of reviews), we would suggest not adding another one. We plan to continue working on dietary aspects in the context of kidney stone prevention and treatment, and in the next article, we will further discuss the pathways for fructose and protein catabolic end products leading to increased oxalate.

Lines 134-148 would be better at the end of the discussion re: the components involved in renal stones so that dietary modifications can be better understood in context. At this point, the text is a broad description that could apply to any disorder where diet has an impact does not provide any specific guidance regarding stone disease.

Response: Thank you for your comment.The fragment was moved to the end of the discussion.

Line 174: Cations does not require an apostrophe.

Response: It has been corrected.

Line 196: Alkalinizing urine is very useful in uric acid stones but not advisable for calcium oxalate stone formers as calcium oxalate becomes less soluble at higher pHs.

Response: Thank you for your comment. This information has been added to the manuscript.Please see lines 191-192.

Line 204: The benefit vs risk of grapefruit juice (as opposed to other citrus fruits) should be noted. Some epidemiological studies show increased risk of stones with grapefruit juice ingestion whereas there are other studies that show no change, improvement or worsening in some parameters associated with stone formation (e.g., increased citrate excretion but increased oxalate excretion also). These discrepancies should be noted.

Response: Thank you for your comment. This information has been added to the article according to comments. Please see lines 199-208.

Lines 217ff:  The intake of protein can have disparate effects on stone formation depending on the type of stone (uric acid vs calcium oxalate or phosphate). These should be distinguished. Also L methionine is not a protein  and should be considered distinct from protein itself in this regard.

Response: Thank you. This paragraph has been expanded and corrected. Please see lines 231-250.

Lines 242-245:  What is the data regarding oral glucose load and calcium handling by the intestine and kidney tubules? No references are given.

Response: I apologise for the oversight. The reference has been added.

Lines246-249:  Ref. #48 does not address fructose effects on calcium, oxalate and urine pH; the study was a euglycemic insulin clamp.

Response: I apologise for the oversight. The reference has been corrected.

Lines 278-289:  Any data on the mechanism whereby a normal calcium diet reduces renal stone formation?

Response: Thank you for your suggestion. The mechanism has been added to the paragraph.

Please see lines 306-313.

Lines 385 – 386: Again, extrapolation that increased calcium excretion (Ref 67) results from decreased tubular reabsorption of calcium without direct data is not accurate. It could well be that alcohol increases calcium filtration (due to lower serum albumin and more free calcium) and thus increases excretion. Important not to extrapolate mechanisms from observations not designed to address the mechanism itself.

Response: Thank you very much for your valuable comment, the fragment has been removed from the text.

Line 403: What is “genetically predicted coffee…consumption”? Is there a gene that makes people drink caffeine (twin studies have been published but the gene is not identified) ? Is caffeine “metabolism” intended via cyp1A2? Likewise “genetically predicted teach intake” (line 443).

Response: Thank you very much for your valuable comment. The issue has been explained, and the text has been appropriately supplemented. Please see lines 430-435 and 474-477

Lines 496ff: Any idea why the conflicting findings with vit C?  Please provide a critical appraisal of the studies that may identify potential reasons for the disparate findings as done for physical activity?

Response: Thank you very much for your valuable comment. The issue has been supplemented. Please see lines 548-559.

Line 536: Unclear why diabetes and obesity are categorized under “disease of affluence”. Poverty also results in diabetes and obesity due to ingestion of cheaper but high calorie, less nutritional foods.  Please recharacterize especially as the subsequent paragraphs actually contradict this heading.

Response: Thank you for your valuable comment. We agree with you. The name of the subsection has been changed to: Disease influenced by lifestyle realated to kidney stone disease

Lines 577: Is the higher urinary excretion of stone forming substances in obese patients due to increased intake rather than the obese status itself or other factor?

Response: Completion has been added to the publication. Please see lines 632-642.

We did our best to improve the manuscript, we hope that we have met your expectations.

Thank you.

Reviewer 3 Report

Comments and Suggestions for Authors

I want to congratulate the authors for a well written review. I truly appreciated the way this review was structured, offering a detaliate and logical description of an ever actual theme. It is a difficult task to organise a large quantity of data.

There are only a few issues that, in my opinion, should be addressed. 

The number of references is impressive. Please describe the reasons why you selected only papers published after 2000. I know that otherwise the number of evaluated papers would have been huge and clearly I understand that you must choose a period to analyse, but I feel that it would be interesting for the readers to understand the reasoning behind this decision. Also, maybe a flowchart detailing the selection process of the analysed papers would be useful.

The phrase on lines 204-205 may be misleading. The grapefruit juice, as an exception among other citric juices, may increase the risk of stone forming. There are a lot of articles detailing this effect. Please review and if necessary discuss.

Author Response

RESPONSE TO REVIEWER 2 COMMENTS

We would like to express our appreciation to the reviewers and editorial board for taking the time and effort to improve our work and provide such insightful comments.

We are pleased to have been given the opportunity to revise our manuscript entitled “The multidisciplinary approach in the management of patients with kidney stone disease – a state-of-the-art review”.

We have carefully reviewed your comments. Below we explain how we revised the paper based on your comments and recommendations.

  1. The number of references is impressive. Please describe the reasons why you selected only papers published after 2000. I know that otherwise the number of evaluated papers would have been huge and clearly I understand that you must choose a period to analyse, but I feel that it would be interesting for the readers to understand the reasoning behind this decision. Also, maybe a flowchart detailing the selection process of the analysed papers would be useful.

Response: Following an initial review, the authors decided to narrow the scope of the literature analysis. The primary reason for this decision was the numerous contradictory reports published over the past several decades. Limiting the database record search to the period from 2000 to 2024 allowed for the presentation of the most current data regarding the issue under study. After a detailed analysis of the selected literature, 135 articles were ultimately included in the review (in the revised version, more studies were included as a result of adding an additional chapter on primary hyperparathyroidism (as recommended by the second reviewer).

  1. The phrase on lines 204-205 may be misleading. The grapefruit juice, as an exception among other citric juices, may increase the risk of stone forming. There are a lot of articles detailing this effect. Please review and if necessary, discuss.

Response: Thank you very much for your comment. We are apologizing for this misleading. This paragraph has been expanded and corrected according to your comment. Please see lines 214-223.

Additionally, following the recommendation of the editorial board, the article has been enriched with one figure (Figure 1). The most important risk factors for kidney stone disease), and a table (Table1) summarizing the current knowledge regarding recommendations for the treatment and prevention of kidney stones.

We did our best to improve the manuscript, we hope that we have met your expectations.

Thank you.

Round 2

Reviewer 3 Report

Comments and Suggestions for Authors

Thank you for answering to all the queries. I believe “ultra-high” temperature milk from line 236 should be removed, as it is already mentioned on line 235.

Author Response

Thank you for your valuable comment and efforts to improve our manuscript.. The“ultra-high” temperature milk from line 236  was removed.